# Influence of Polymer Composition on the Controlled Release of Docetaxel: A Comparison of Non-Degradable Polymer Films for Oesophageal Drug-Eluting Stents

**DOI:** 10.3390/pharmaceutics12050444

**Published:** 2020-05-11

**Authors:** Paris Fouladian, Franklin Afinjuomo, Mohammad Arafat, Amanda Bergamin, Yunmei Song, Anton Blencowe, Sanjay Garg

**Affiliations:** 1Pharmaceutical Innovation and Development (PIDG) Group, Clinical and Health Sciences, University of South Australia, Adelaide, SA 5000, Australia; Paris.fouladian@mymail.unisa.edu.au (P.F.); olumide.afinjuomo@mymail.unisa.edu.au (F.A.); mohammad.arafat@mymail.unisa.edu.au (M.A.); bergamin.amanda@gmail.com (A.B.); May.Song@unisa.edu.au (Y.S.); 2Applied Chemistry and Translational Biomaterials (ACTB) Group, Clinical and Health Sciences, University of South Australia, Adelaide, SA 5000, Australia

**Keywords:** docetaxel, non-degradable polymer, controlled release, oesophageal drug-eluting stent

## Abstract

Following the huge clinical success of drug-eluting vascular stents, there is a significant interest in the development of drug-eluting stents for other applications, such as the treatment of gastrointestinal (GI) cancers. Central to this process is understanding how particular drugs are released from stent coatings, which to a large extent is controlled by drug-polymer interactions. Therefore, in this study we investigated the release of docetaxel (DTX) from a selection of non-degradable polymer films. DTX-polymer films were prepared at various loadings (1, 5 and 10% *w*/*w*) using three commercially available polymers including poly(dimethylsiloxane) (PSi), poly (ethylene-*co*-vinyl acetate) (PEVA) and Chronosil polyurethane (PU). The formulations were characterised using different techniques such as photoacoustic Fourier-transform infrared (PA-FTIR) spectrophotometry, X-ray diffraction (XRD) and differential scanning calorimetry (DSC). The effect of DTX on the mechanical properties of the films, in-vitro release, and degradation tests were also assessed. For all polymers and DTX loadings, the drug was found to disperse homogenously without crystallisation within the polymer matrix. While no specific interactions were observed between DTX and PSi or PEVA, hydrogen-bonding appeared to be present between DTX and PU, which resulted in a concentration-dependent decrease in the Young’s moduli of the films due to disruption of inter-polymeric molecular interactions. In addition, the DTX-PU interactions were found to modulate drug release, providing near-linear release over 30 days, which was accompanied by a significant reduction in degradation products. The results indicate that DTX-loaded PU films are excellent candidates for drug-eluting stents for the treatment of oesophageal cancer.

## 1. Introduction

A recent report from the Global Burden of Disease Cancer Collaboration highlighted oesophageal cancer as the 11th most common type of cancer by the number of incidents in both sexes globally [1]. Unfortunately, at the time of diagnosis, most patients are in advanced stages of the disease and palliative action remains the only treatment option for many patients [2,3]. The most prominent symptom that causes difficulties in patients is dysphagia, which causes excessive secretions, trouble in swallowing foods and weight loss, that negatively impact the patient’s quality of life [4]. Palliative treatment with stents is commonly used and proved to be effective in oesophagus luminal patency and morbidity, and mortality reduction [5,6]. Although stent placement can provide rapid and effective action, the recurrence of the lumen due to ingrowth and overgrowth within the wall of the oesophagus leads to narrowing of the lumen and shortening the effective life expectancy of the treatment [7,8].

In the past few decades, several attempts have been made to develop drug-eluting stents (DESs) as an effective choice for the treatment of occlusion or stenosis of the body’s tubular structures [9,10]. DESs have shown a significant function by providing both mechanical support and release of drugs to prolong palliation in the malignant tumours [10,11]. For many DESs, a drug-loaded polymer membrane acts as a drug reservoir and provides sustained drug release [12]. Thus, the combination of drug, polymer and stent in the form of a DES can be included in a controlled therapeutic protocol as a means to enhance the therapeutic concentrations of the chemotherapeutic agents, to prolong the release of the drugs, to avoid wastage of drugs and to reduce systemic side effects [13].

Polymeric materials are used extensively for implantable devices and play a crucial role in delivering therapeutic agents to targeted tissues. A wide range of polymeric materials have been investigated as coatings and membranes for DESs and can be divided into degradable and non-degradable polymers. For the former, degradation of the polymer coating may result in stent migration, stricture recurrence or stent obstruction due to malignant tumour ingrowth, and therefore, non-degradable polymers are preferred for DESs [14]. Frequently used polymeric materials for non-vascular stents and DES coatings include polyurethane, poly(tetrafluoroethylene) (PTFE), poly(ethylene terephthalate) (PET) and polysiloxanes [14,15,16].

Polysiloxanes are some of the most extensively used materials in pharmaceutical industry. Because of their good thermal and oxidative stability, low toxicity, compatibility and excellent resistance to irradiation degradation, they are well suited for the fabrication of medical devices such as catheters, stents, drains, cosmetic implants and cardiac leads [17]. Polysiloxanes are also widely used as excipients in topical formulations [18], and transdermal patches as active ingredients in anti-flatulence medication, as well as controlled delivery platforms [19,20]

Polyurethanes are also commonly used biomedical materials and because of their structural diversity and exceptional elasticity, durability, compliance, fatigue resistance and biocompatibility have been employed in the development of biomedical devices, artificial organs, tissue engineering scaffolds and drug delivery systems [21]. Polyurethanes are a class of segmented copolymers consisting of soft and hard segments; the hard domains are responsible for mechanical strength, whereas the soft segments provide flexibility [22]. Polyurethanes have previously been applied as controlled drug delivery systems in vaginal rings [23], implants [24], coatings [25] and stents [26,27].

Poly(ethylene-*co*-vinyl acetate) (PEVA) is one of the most commonly used polymers in implantable and transdermal controlled released applications. PEVA is a nondegradable and non-toxic thermoplastic copolymer consisting of ethylene and vinyl acetate (VA), with VA content generally ranging between 0 and 40 wt % [28]. In the biomedical field, PEVA has found widespread commercial application as a sustained release platform, including intravaginal rings (NuvaRing^®^) [29], intrauterine systems (Progestasert^®^) [30], subcutaneous implants (Implanon^®^) [31], stents and coating systems (Cypher™) [32].

Several preclinical [33,34,35,36,37,38,39,40] and clinical studies [41,42,43] have used polyurethanes, silicones and PEVA as carriers for drug delivery in gastrointestinal malignancies. However, limited studies about localised drug delivery, particularly using these materials to oesophageal cancer have been conducted so far. In a blinded and randomised clinical study conducted in 21 patients with oesophageal cancer, Manifold et al. [44] incorporated paclitaxel into PEVA (33% *w*/*w*) coated stents. Whilst no significant differences between the paclitaxel eluting stent and uncoated control stents were found, this study provided the basis for further testing of DESs in oesophageal cancer clinical studies. Dai et al. also confirmed the safety and efficacy of iodine eluting polyurethane coated stents for the treatment of malignant oesophageal strictures [45]. Whilst the candidate polymers for our study have been commonly used for DES, it is interesting to note that no comparative investigations assessing the physicochemical and release properties of different polymers have been conducted in a single study.

Docetaxel (DTX) is a cytotoxic antineoplastic agent with high potency and activity against various common cancers, as demonstrated in animal models and clinical studies [46,47]. However, DTX is also highly cytotoxic to healthy tissue and has a low aqueous solubility. Thus, the only commercial product containing DTX is an intravenous dosage formulation containing ethanol and polysorbate 80. Resultingly, DTX therapy is associated with systemic side effects such as neurotoxicity, musculoskeletal toxicity and neutropenia, cumulative fluid retention and peripheral neuropathy caused by prolonged infusion of DTX, and also severe hypersensitivity reactions and hyperlipidemia caused by polysorbate 80. Considering these drawbacks, recent and extensive effort has been focused on the development of less toxic formulations that avoid the use of polysorbate 80 and provide more targeted drug delivery [48]. For example, several targeted drug delivery systems including nanoparticles [49], micelles [50], liposomes [51] and prodrugs [52] have been developed. In addition to these systems, the development of a localized DTX delivery systems would be highly advantageous to reduce systemic toxicity, improve patient compliance and improve therapeutic outcomes [53,54,55].

DESs provide a unique opportunity to deliver therapeutics, such as DTX, locally from a polymer coating to the target tissue, whilst minimising systemic toxicities. As with any polymer-based drug-eluting system, careful selection of the polymeric material is required to ensure optimal drug-polymer interactions and release characteristics. Therefore, in this study we assessed a selection of clinically relevant non-biodegradable polymers (poly(dimethylsiloxane) (PSi), PEVA and Chronosil polyurethane (PU)) to prepare DTX-loaded films as potential DES covering materials for localized delivery of DTX to the oesophagus lumen. The performance of the different polymer compositions was determined using various analytical techniques to allow selection of the most appropriate DTX-polymer combination based on physical characteristics and release behaviour.

## 2. Materials and Methods

### 2.1. Materials:

DTX (>99%) was purchased from Shanghai Jinhe Bio-Technology Co., Ltd. (Shanghai, China). ChronoSil AL 80A 5% Silicone was purchased from AdvanSource Biomaterials (Wilmington, MA, USA). Slygard^®^186 silicone elastomer kit (Dow Corning) was obtained from C.B.C (Adelaide, SA, Australia). PEVA copolymer with 40% (*w*/*w*) VA content was purchased from Sigma-Aldrich (Castle Hill, NSW, Australia). Tetrahydrofuran (THF), acetic acid, sodium hydroxide and ammonium acetate were purchased from Chem-Supply (Adelaide, SA, Australia) and dichloromethane (DCM) was obtained from Thermo Scientific (Scoresby, Vic, Australia). Acetonitrile (HPLC grade) and Tween-80 were purchased from Merck Pty Ltd (Baywaters, Victoria, Australia). All reagents in this study were of analytical grade and were used without further purification.

### 2.2. Preparation of DTX-Loaded PSi Films

Sylgard^®^186 silicone elastomer kit consists of two parts; elastomer base and a curing agent. Blank (without DTX) and DTX-loaded PSi films were prepared using a casting method. Initially, the elastomer base and curing agent were mixed in a 10:1 volumetric ratio in THF (4 mL). For blank films (PSi), the solution was then cast on to a Petri dish (30 mm diameter) and the solvent was evaporated at 60 °C in an oven for 24 h to provide films with an average thickness of approximately 300 μm. DTX-loaded films were prepared at three different concentrations of DTX with the silicone components to provide final loadings of 1, 5 and 10% *w*/*w* (Psi_1_, Psi_5_ and Psi_10_, respectively); refer to the Appendix A for the actual amount of DTX and polymer used in each formulation (Appendix A). DTX-loaded films were prepared as described for the blank films, with the exception that DTX dissolved in THF (2 mL) was added to the polymer solution with sonication prior to casting.

### 2.3. Preparation of DTX-Loaded PEVA and PU Films

Blank and DTX-loaded PU or PEVA films were prepared via a solvent casting method. For blank films, PU or PEVA were dissolved in THF (4 mL) in a sealed vial and heated in the oven at 40 °C for 24 h to ensure complete dissolution. The solutions were cast into a Petri dish (30 mm diameter), and the solvent was evaporated in an oven at 40 °C for 1 h, followed by 60 °C for 24 h to afford films with an average thickness of approximately 300 μm. DTX-loaded films were prepared at three different concentrations of DTX to achieve loadings of 1, 5 and 10% *w*/*w* (PU_1_, PU_5_ and PU_10_, and PEVA_1_, PEVA_5_ and PEVA_10_) (Appendix A). DTX-loaded films were prepared as described for the blank films, with the exception that DTX dissolved in THF (2 mL) was added to the polymer solution prior to casting.

### 2.4. Photoacoustic Fourier-Transform Infrared (PA-FTIR) Spectroscopy of Films

For characterizing the interaction between the polymer and DTX, PA-FTIR spectroscopy was performed on the films using a Thermo Nicolet Magna 750 FTIR spectrometer (Thermo Nicolet, WI, USA). Spectra were recorded in the spectral range from 400 to 4000 cm^−1^. The spectra were obtained using 256 scans under a helium purge and mirror velocity of 0.158 cm/s.

### 2.5. X-Ray Powder Diffraction (XRD) of Films

XRD was performed on samples using a Malvern Panalytical XRD (Empyrean XRD, Worcestershire, UK) with a Cu Kα radiation source (λ = 0.15406 nm) operating at 40 kV and 30 mA. A two theta (2θ) continuous scan was recorded at a step size of 0.02° from 5° to 40° with a total time per step of 29.07 s.

### 2.6. Scanning Electron Microscopy (SEM) of Films

The surface morphology of the blank and DTX-loaded (10% *w*/*w*) films was investigated using a Zeiss Merlin field emission gun scanning electron microscope (Jena, Germany). The samples were mounted on sample holders using double-sided adhesive tape and after sputter coating with platinum, examined by SEM. The photomicrographs were recorded at an accelerating voltage of 2 KV.

### 2.7. Thermal Analysis of Films

The thermal properties of the films were investigated using thermogravimetric analysis (TGA) and differential scanning calorimetry (DSC). DSC was performed on a Discovery 2920 DSC (TA Instruments, New Castle, DE, USA). Approximately 2 mg of the sample (DTX powder, blank films or DTX-loaded (10% *w*/*w*) films) was placed in the bottom of a hermetic aluminium pan (closed crimped cells), sealed and analysed with a heating rate of 10 °C/min from 25 to 200 °C under nitrogen atmosphere. TGA measurements were performed on a Discovery TGA 550 thermogravimetric analyzer (TA Instruments). Approximately 7 mg of the sample (DTX powder, blank films or DTX-loaded (10% *w*/*w*) films) was heated at a rate of 10 °C/min under a flow of nitrogen from room temperature to 500 °C.

### 2.8. Mechanical Properties of the Films

The tensile properties of the films were assessed using a mechanical tester (Shimadzu, EZ-LX, Kyoto, Japan) fitted with a 500 N load cell. The films were cut into strips (40 mm length, 10 mm width) and then stretched at a rate of 5 mm/s until breakage. The tensile properties of the films were calculated from the resulting force versus displacement curves and are reported as the average of three replicates.

### 2.9. HPLC Analysis of DTX

HPLC was conducted on a Shimadzu system consisting of a DGU-20AS degasser, SIL-20A HT autosampler, and a SPD-M20A UV–vis detector set at a wavelength of 230 nm. The system was fitted with an Eclipse XDB C18 column (4.6 × 150 mm, 3.5 μm particle size) and the oven temperature was set to 25 °C. The mobile phase consisted of ammonium acetate buffer (0.02 M, pH 5) and acetonitrile (43:57% *v*/*v*) at a flow rate of 1 mL/min, and the injection volume was 20 μL.

### 2.10. Determination of Drug-Loading in Films

Because of the different chemical structures of the polymers, two different extraction methods were employed for the determination of DTX-loading in the films.

Determination of DTX in PSi films: Determination of drug loading was conducted as described previously by Shaikh et al. [56] with some modifications. Films were cut to a standard size disc using a punch (11 mm diameter), weighed and then cut into smaller strips. For each disc, the strips were extracted three times using DCM (4 mL) for 4 h, whilst sonicating. The solutions from each extraction were combined, evaporated under a flow of nitrogen, reconstituted in acetonitrile: water (1:1), filtered through a 0.45 µm PVDF filter and analysed via HPLC.

Determination of DTX in PEVA/PU films: Films were cut into 11 mm diameter discs using a punch and weighed. Each disc was cut into 5-6 small strips and dissolved in THF (1 mL). The solution was then added dropwise to a stirred solution of ammonium acetate buffer (0.02 M, 25 mL, pH 5) and acetonitrile (43:57% *v*/*v*). The samples were sonicated for 30 min, filtered through a 0.45 µm PVDF filter and analysed via HPLC.

### 2.11. Determination of DTX Solubility

The solubility of DTX in the media used for release studies was determined by adding DTX (1 mg) to the release media (0.1 M phosphate buffer at pH 6.5 containing 0.1% *v/v* Tween 80, 5 mL) and the mixture was shaken at 40 rpm and 37 °C for 48 h. The mixture was then centrifuged at 13,000 rpm for 5 min and the supernatant was passed through a 0.45 μm membrane filter and the concentration of DTX was determined by HPLC. This procedure was repeated in triplicate. The solubility of DTX in the release media was determined to be 10.26 ± 1.72 μg/mL.

### 2.12. In Vitro Drug Release

The release of DTX from the polymer films into media was conducted in triplicate over a period of 31 d. The DTX-loaded films (5% *w*/*w*) were selected for in vitro release studies. The films were punched into discs with a diameter of 8 mm and then placed in release media (0.1 M phosphate buffer at pH 6.5 containing 0.1% *v/v* Tween 80) with shaking at 40 rpm in an incubator at 37 °C. Based on the different rates of release observed in a preliminary study (data not shown), the PEVA discs were placed in vials containing 40 mL of release media, the PU discs in 20 mL of release media and the PSi discs in 10 mL of release media. Samples were collected at predetermined time points and to maintain sink conditions all the media was replaced with fresh media at each time point. The amount of the released DTX and relative concentration of degradation products were measured via HPLC.

## 3. Results

### 3.1. Preparation of DTX-Loaded Films

All films were prepared using a casting/solvent evaporation method. For each polymer, three DTX-loadings (1, 5 and 10% *w*/*w*) were studied to evaluate and compare the physicochemical properties and drug release characteristics of the films with different loadings. Whilst optimal intravenous doses of DTX have been determined through clinical studies [57], the concentration required for a localised delivery system still requires extensive in vivo testing, and therefore, a range of loadings were evaluated to provide an indication of how drug-loading effects the performance. This would provide a foundation for determination of an optimal formulation in subsequent in vivo and clinical trials.

Three non-degradable polymers were selected in this study to represent commonly used biomedical materials for implants and drug-delivery systems, including PEVA (40% *w*/*w* vinyl acetate (VA) content), PU (ChronoSil AL 80A, 5% Silicone) and PSi (Sylgard^®^186, a two-component poly(dimethyl siloxane) (PDMS) elastomer) [58]. Whilst the PEVA, PEVA_1_ and PEVA_5_ films were transparent, the PEVA_10_ film was opaque, possibly because of drug aggregation and crystallisation. Nevertheless, all PEVA films were flexible with thicknesses of 304 ± 2.23 µm. All blank and DTX-loaded PU films were opaque with a thickness of 304 ± 7.56 µm, whereas all PSi films were a translucent white with thicknesses of 314 ± 12.3 µm.

### 3.2. PA-FTIR Spectrophotometry of DTX-Loaded Films

To assess the distribution of DTX within the polymer films and possible interactions with the polymers, PA-FTIR spectroscopy was conducted on pure DTX, and the blank and DTX-loaded (10% *w*/*w*) polymer films (Figure 1). The spectrum of DTX revealed strong absorbances at 1500, 1720, 2800–3100 and 2800–3100 cm^−1^ consistent with aromatic C=C, C=O, C–H and O–H/N–H stretches, respectively. The blank polymer films also showed characteristic peaks consistent with their structure. For example, the spectra for PEVA and PU displayed peaks at approximately 1250, 1740 and 2800–3000 cm^−1^ corresponding to C–O, C=O and C–H stretches, whereas PSi peaks were observed at approximately 800–900, 1000–1150 and 3000 cm^−1^ consistent with Si–C, Si–O and C–H stretches, respectively. In all cases, the spectra of the drug-loaded polymer films looked similar to their blank counterparts and there were no obvious peaks observed corresponding to DTX, probably as a result of the relatively low DTX concentration and overlap of peaks between the drug and polymers. A slight broadening of the peak at 3300–3500 cm^−1^ observed in the spectrum of the DTX-loaded PU film may be an indication of hydrogen-bonding between the drug and polymer [53].

### 3.3. XRD of DTX-Loaded Films

XRD was used to investigate the possible crystallization of DTX within the drug-loaded films. The crystalline nature of pure DTX displays several distinct 2θ peaks between 7 and 23.5°, as previously reported in the literature [53] (Figure 2A). In comparison, none of the drug-loaded films displayed any sharp peaks corresponding to DTX, which may indicate that DTX is molecularly dispersed or is in an amorphous form and any crystalline component is below the sensitivity of detection (Figure 2B–C). Rather, all of the DTX-loaded polymer films displayed broad peaks. For example, the blank and DTX-loaded PEVA films displayed broad 2θ peak at ~20° consistent with the amorphous nature of PEVA [59,60]. Blank PU film displayed broad 2θ peaks at approximately 11 and 20.0° (Figure 2C), assigned to scattering from PU chains with regular interplanar spacing. However, the inclusion of DTX resulted in a noticeable decrease in the 2θ peak at 11°, which has previously been reported to result from rigid polyurethane segments [61]. Thus, the DTX appears to interfere with the interactions between the polyurethane segments, reducing the overall crystallinity of the polymer. For the PSi films, two broad 2θ peaks were observed at approximately 12 and 22° consistent with PDMS (Figure 2D) [62]. These did not change significantly upon DTX loading suggesting that DTX does not interfere with the polymer morphology as was the case for the PEVA films.

### 3.4. Surface Topography of the DTX-Loaded Films

The morphology of DTX powder, blank and DTX-loaded films (10% *w*/*w*) were characterized using SEM (Appendix A). DTX powder presented with characteristic irregular needle-shaped crystal morphologies consistent with a low molecular weight crystalline substance. In contrast, all DTX-loaded films displayed flat and smooth topographies (Appendix A), and as expected, the DTX crystals were not observed in any of the polymer films, indicating incorporation of the DTX within the films. SEM images were also recorded for the blank polymeric films. Comparison of the blank and DTX-loaded films revealed that there were no significant differences in the topography of the films. The SEM images confirm that the DTX was successfully loaded into the films and that the topography of the films is not influenced by DTX.

### 3.5. Thermal Analysis of DTX-Loaded Films

DSC was conducted to investigate the distribution of DTX within the polymer films and the effect of the drug on the thermal phase transitions of the polymers (Figure 3). As expected, the thermogram of pure DTX was characterized by an endothermic peak at 169 °C corresponding to the melting temperature (*T_m_*) [63]. In comparison, the absence of a DTX *T_m_* in the thermograms of the DTX-loaded films (10% *w*/*w*) suggested DTX was not predominately present in a crystalline form, corroborating the XRD results (Figure 2). The absence of a DTX *T_m_* may be due to the dissolution of the drug in the polymer matrix when it melts, or the presence of amorphous drug aggregates that lack a distinct, sharp melting point. For the blank and DTX-loaded PEVA films an endothermic peak was observed in the thermogram at ~46 °C corresponding to the *T_m_* for PEVA (40% VA) [32,64], and indicate that the inclusion of DTX does not influence the morphology of the polymer nor does the drug interact strongly with the polymer. The thermogram of blank PU displayed glass transition temperatures (*T_g_*) at 70 and 92 °C that correspond to different segments of the PU [65]. Upon the incorporation of DTX into the PU film, the *T_g_* at 92 °C is absent indicating that the drug may be preferentially interacting with rigid PU segments of the polymer and preventing it from forming interactions between these domains. This observation is also consistent with the XRD results for the PU films, which revealed the disappearance of a 2θ peak at 11° upon the inclusion of DTX (Figure 3C). The thermograms of the blank and DTX-loaded PSi films did not display any distinct events across the temperature range studied, consistent with cross-linked polysiloxanes [56].

TGA analysis was conducted to determine the thermal stability and decomposition temperature of DTX within the polymer films. TGA thermograms and derivative thermogravimetric (DTG) curves are shown in Figure 4A,B, respectively. Whilst pure DTX displayed a sharp weight loss and degradation event at ~220 °C, followed by continuous and gradual weight loss, the blank PEVA, PU and PSi films were thermally stable until ~320, 240 and 360 °C, respectively. The DTG thermogram for the PEVA film revealed two distinct degradation events with degradation peak temperatures (*T_d_*) of 360 °C (onset ~300 °C) and 470 °C, corresponding to decarboxylation of the VA component and depolymerisation, respectively [66]. Similarly, the thermogram of the PU film displayed a two-stage degradation process as reported previously in the literature [67]. The first stage, ranges from 230 to 280 °C with a weight loss of ~10%, which is then followed by a significant weight loss with a *T_d_* of 360 °C. Previously, it has been reported that the first and second weight loss events result from degradation of the hard and soft segments of PUs, respectively [67,68,69]. In comparison, PSi was characterized by a high degradation temperature with only a gradual weight loss above 380 °C, and among the three polymers examined, displayed the highest thermal stability characteristic of polysiloxanes.

Compared to the blank PEVA film, the thermogram of the DTX-loaded PEVA_10_ film displayed an additional *T_d_* at 260 °C resulting from DTX degradation, and implying that the polymer matrix provides some thermal protection. Surprisingly, the incorporation of DTX into the PU_10_ and PSi_10_ films was found to slightly increase the thermal stability of the formulations. In all cases, the incorporation of DTX into the polymer films was found to improve the thermal stability of the drug.

### 3.6. Mechanical Properties

The tensile properties of the blank and DTX-loaded films (1, 5 and 10% *w*/*w*) were investigated to determine the effect of the drug on the mechanical characteristics of the polymer films. Stress–strain curves (Figure 5) were recorded until breakage and the ultimate tensile strength, elongation at break, toughness and stiffness (Young’s moduli) were recorded (Table 1). Comparing the blank films, it was evident that PEVA films showed the highest elongation at break, PU films had the highest toughness and stiffness, and PSi films displayed the lowest elongation at break and toughness. The incorporation of DTX into each polymer film exhibited markedly different effects on the mechanical properties.

The incorporation of DTX into the PEVA films at all loadings had little effect on the profile of the stress–strain curve, although the elongation at break and ultimate tensile strength were found to decrease slightly leading to an overall reduction in toughness with increasing loading. Interestingly, the incorporation DTX into the PU films was not found to noticeably change the elongation at break, although higher DTX-loadings (5 and 10% *w*/*w*) lead to a significant decrease in the ultimate tensile strength, toughness and stiffness. For the PSi films, the inclusion of DTX lead to a significant decrease in all measured parameters, even at low drug loadings (i.e., 1% *w*/*w*).

The results indicate that the incorporation of DTX into PEVA films has minimal effect on their mechanical behaviour in the elastic region and maybe a good choice when DTX-loaded films with identical mechanical properties to the blank film are required. The constant elongation at break, and reduction in toughness and stiffness for the PU films upon DTX loading may be useful when more flexible film is required. Interestingly, the significant reduction in the toughness and stiffness of the PU_5_ and PU_10_ adds further support to the hypothesis that DTX interferes with the H-bonding interactions between the urethane segments of the polymer, which would be responsible for imparting toughness [61]. The significant decrease in the mechanical properties for the PSi films upon DTX loading may result from the DTX interfering with the cross-linking reaction used to form the silicone matrix and may be a problem if the film needs to be stretched significantly.

### 3.7. Determination of DTX Loading

The DTX loading of the polymeric films (1, 5 and 10% *w*/*w* of DTX) was investigated to determine the absolute amount of extractable drug as compared to the theoretical loading, assess whether the DTX was homogeneously dispersed within the films, and to determine if the processing conditions for film formation resulted in degradation of the DTX. For all of the DTX-loaded PEVA, PU and PSi films the experimental loading (i.e., extractable DTX) was found to be similar to the theoretical loadings (based on the amount of DTX used to prepare the films), providing percentage recoveries of >91% (Table 2). The reduction in recovery percentage (<100%) could be related to degradation of DTX during film processing or extraction as well as incomplete extraction of DTX due to interactions with the hydrophobic polymers.

### 3.8. In Vitro Drug Release

The release profile of DTX from the DTX-loaded films (5% *w*/*w*) was studied in vitro by submerging the films in phosphate buffer at pH 6.5 containing 0.1% *v/v* Tween 80 and monitoring the concentration of DTX using HPLC over 31 d. The media was completely changed every two days to prevent saturation of DTX in the receiving solution (solubility of DTX in the media was determined to be 10.26 ± 1.72 μg/mL). Both DTX-loaded PEVA_5_ and PU_5_ films displayed an initial burst release (44 and 13%, respectively) after 24 h, which was then followed by a sustained, near-linear release of DTX (Figure 6). Similarly, the PSi film also displayed a near-linear release, although no noticeable burst release was detected. Over 31 d the total release of DTX from the PEVA_5_, PU_5_ and PSi_5_ films was determined to be 63, 48 and 10%, respectively.

PEVA drug-loaded film exhibited the highest release rate among the three polymers due to the weak interactions between DTX and PEVA. This assumption was further confirmed with nearly complete release of DTX from PEVA_1_ films in 9 days (Appendix A). DTX-loaded PSi_5_ films showed the lowest rate of the release among the films, which may result from the highly cross-linked matrix of PSi_5_ that restricts and slows drug diffusion.

These results show a different rate of sustained release for all the polymers after the initial burst release and imply that changing the polymer composition is an effective method for controlling the drug release profile, which may be useful for different clinical scenarios. For instance, PEVA_5_ films provide a rapid release followed by a relatively slow zero-order release, whereas PSi_5_ films provide a similar zero-order release without the initial burst release. In comparison, PU_5_ films provide a slight burst release but are predominately characterized by a more rapid release over a sustained period. The initial burst release may be useful in reducing the tumour burden while the sustained release will be necessary to suppress any growth from the remnant cancer cells. The release profiles for 1 and 10% drug loaded films are provided in the Appendix A.

### 3.9. Drug Release Mechanism

To assess the different types of release profiles from the films, the in vitro drug release data for the DTX-loaded PEVA_5_, PU_5_ and PSi_5_ films (5% *w*/*w*) was fitted to the various mathematical models, including the Higuchi, First-order, Zero-order, Hixon-Crowell and Korsmeyer–Peppas models (Appendix A). For PEVA_5_, PU_5_ and PSi_5_ films, the highest correlation coefficient (R^2^) value was used to determine the most suitable mathematical model (Table 3). For both PU_5_ and PSi_5_ films the experimental release kinetics were best fitted by the Higuchi model with R^2^ values of 0.9950 and 0.9922, respectively, whereas the Kors–Peppas model fitted best with the PEVA_5_ release data (R^2^ = 0.966). To study the release mechanism, the experimental release data for the DTX-loaded films was plotted as the log cumulative % drug release versus log time to determine the diffusion exponent (*n*) [70]. All of the polymer films investigated displayed n values of <0.5 indicating a diffusion-controlled drug release mechanism, which is consistent with previous reports of drug release from PU, PEVA and PSi systems [71,72]

### 3.10. Degradation Studies

The degradation of DTX during its release from polymer matrices is an important consideration as it may lead to sub-optimal dosing. In this study, we were interested to understand if the composition of the polymer matrix influences the stability of DTX and the formation of degradation products. As reported previously by Kumar et al. [73] and supported by previous work in our laboratory [56], DTX can hydrolyse, oxidise and epimerise under acidic and basic conditions to afford various derivatives including 10-deacetyl baccatin III, 7-epi-10-deacetyl baccatin III, 7-epi-10-oxo-10-deacetyl baccatin III, 7-epi-DTX and 7-epi-10-oxo-DTX. In this study we focused on the two main degradants observed upon release of DTX from the polymer films, namely 7-epi-DTX and 7-epi 10-oxo-DTX (Appendix A), which were quantified concurrently with DTX release and are reported as the relative percentages (Figure 7; DTX + 7-epi-DTX + 7-epi-10-oxo-DTX = 100%).

All three drug-loaded films showed evidence for the formation of 7-epi-DTX and 7-epi-10-oxo-DTX, although to different extents. For the DTX-loaded PEVA_5_ film, both 7-epi-DTX and 7-epi-10-oxo-DTX were observed at each timepoint and after the initial 7 days, the amount of each remained high at between 4–7 and 12–16%, respectively. Whilst the trend observed for the formation of 7-epi-DTX in the DTX-loaded PU_5_ films was similar to the PEVA films, the relative amounts were reduced by approximately half. In addition, the 7-epi-10-oxo-DTX derivative was predominately observed over the first 11 days. The differences noted between the PEVA and PU films suggest that secondary interactions between the DTX and PU films help to stabilize DTX and prevent epimerization and oxidation. This may include H-bonding between the DTX and the urethane linkages, which was indicated in the FTIR spectroscopy results (Figure 1B). For the DTX-loaded PSi films, there were no obvious trends with the concentration of the 7-epi-DTX and 7-epi-10-oxo-DTX derivatives ranging from 4–15 and 0–14%, respectively.

Overall, there was a decline in the native form of DTX for all the polymer films with different rates and slopes. It is worthwhile to mention that the rate of 7-epi-DTX was significantly higher than 7-epi-10-oxo in all three polymer films. Furthermore, PEVA and PSi showed the highest decline in the native form of DTX as well as the highest rates of both degradants. On the other hand, PU films showed a gradual decrease in the native form of DTX with a consistent rate. No clear reason can be attributed to the different percentages of the degradants at different sampling time points for each polymer, although further studies are underway to gain more insights into the reason behind the different rates of degradation with the different polymers.

## 4. Conclusions

A series of clinically relevant non-degradable polymers were investigated as drug eluting polymer films that could potentially be incorporated into for DTX-loaded stent coatings. All of the polymers were physically and chemically compatible with DTX and showed a sustained release of DTX for more than 30 days. For all polymers and DTX loadings, the drug was found to disperse homogenously without crystallisation within the polymer matrix. In particular, the DTX-polyurethane interactions were found to modulate drug release, providing near-linear release over 30 days, which was accompanied by a significant reduction in degradation products. These results indicate that DTX-loaded polyurethane films are excellent candidates for drug-eluting stents for the treatment of oesophageal cancer. Nevertheless, it is important to note that further studies are required to optimise the films for drug-eluting stents, and in particular the dose and duration that might be required in treatment regimes.

## Figures and Tables

**Figure 1 pharmaceutics-12-00444-f001:**
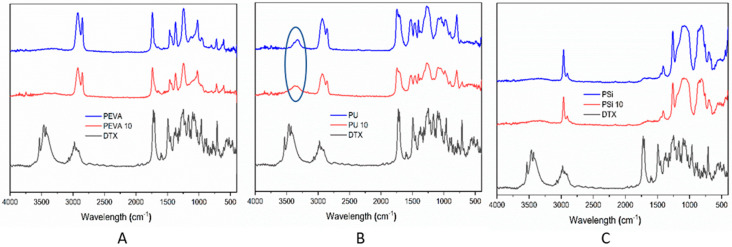
(**A**) Photoacoustic Fourier-transform infrared (PA-FTIR) spectra of docetaxel (DTX), blank PEVA film (PEVA) and DTX-loaded PEVA film (10% *w*/*w*; PEVA_10_). (**B**) PA-FTIR spectra of DTX, blank polyurethane (PU) film (PU) and DTX-loaded PU film (10% *w*/*w*; PU_10_); circled region highlighting differences because of possible drug-polymer hydrogen-bonding interactions. (**C**) PA-FTIR spectra of DTX, blank silicone film (PSi) and DTX-loaded PSi film (10% *w*/*w*; PSi_10_).

**Figure 2 pharmaceutics-12-00444-f002:**
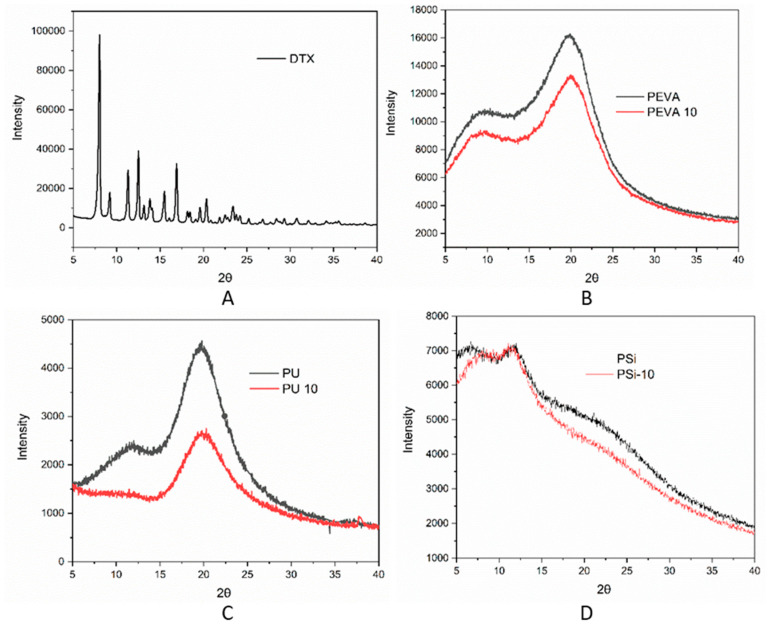
X-ray diffractograms of (**A**) DTX powder, (**B**) blank PEVA (PEVA) and 10% *w*/*w* DTX-loaded PEVA film, PEVA_10_ (red), (**C**) blank PU(Black) and 10% *w*/*w* DTX-loaded PU film, PU_10_ (red), and (**D**) blank PSi (black) and 10% *w*/*w* DTX-loaded PSi film, PSi_10_ (red).

**Figure 3 pharmaceutics-12-00444-f003:**
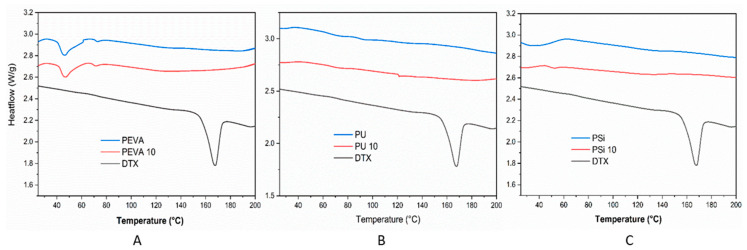
Differential scanning calorimetry (DSC) thermograms of (**A**) DTX, blank PEVA (PEVA) and 10% *w*/*w* DTX-loaded PEVA film (PEVA_10_), (**B**) DTX, blank PU (PU) and 10% *w*/*w* DTX-loaded PU film (PU_10_), and (**C**) DTX, blank Psi (PSi) and 10% *w*/*w* DTX-loaded PSi film (PSi_10_).

**Figure 4 pharmaceutics-12-00444-f004:**
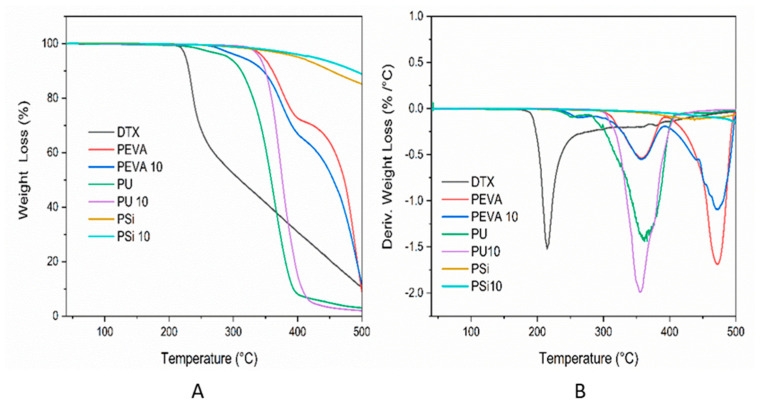
(**A**) Thermogravimetric analysis (TGA) thermograms and (**B**) derivative thermogravimetric (DTG) curves of DTX, PEVA, PU, Psi and 10% *w*/*w* DTX-loaded PEVA10, PU10, Psi10 films.

**Figure 5 pharmaceutics-12-00444-f005:**
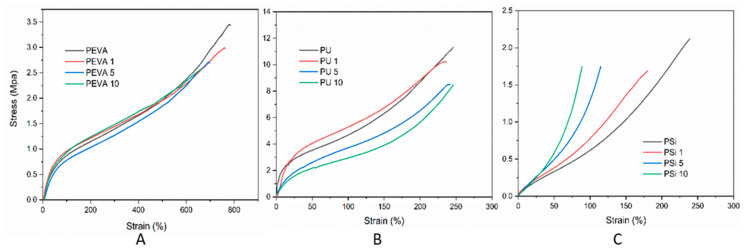
Stress–strain curves of blank, 1, 5 and 10% *w*/*w* DTX-loaded polymer films: (**A**) PEVA, PEVA_1_, PEVA_5_ and PEVA_10_, (**B**) PU, PU_1_, PU_5_ and PU_10_ and (**C**) PSi, PSi_1_, PSi_5_ and PSi_10_.

**Figure 6 pharmaceutics-12-00444-f006:**
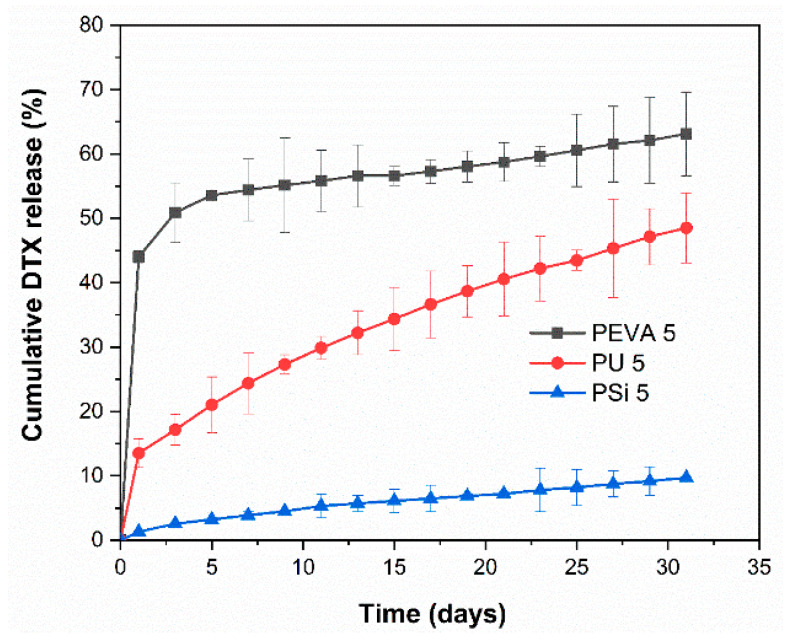
In vitro DTX release profiles from the PEVA_5_, PU_5_ and PSi_5_ films over 31 days in phosphate buffer at pH 6.5 containing 0.1% *v/v* Tween 80.

**Figure 7 pharmaceutics-12-00444-f007:**
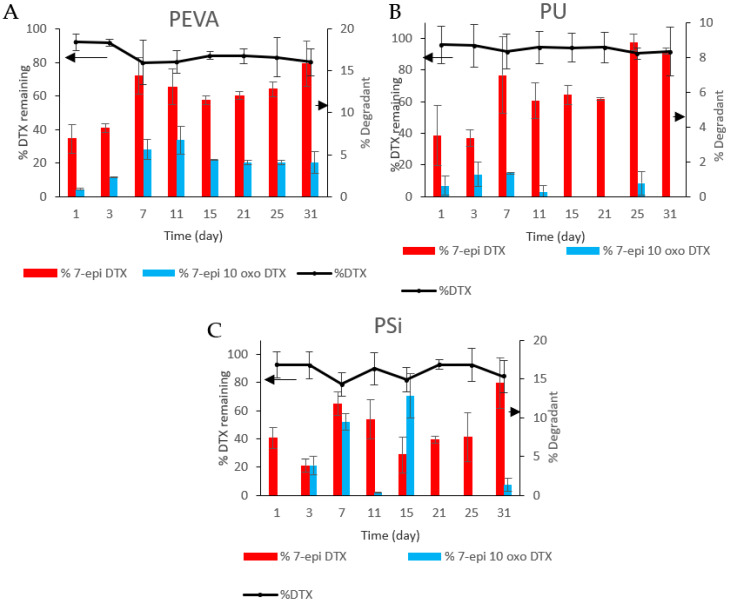
Percentage DTX remaining and percentage release of 7-epi-DTX and 7-epi-10-oxo DTX from (**A**) PEVA_5_, (**B**) PU_5_ and (**C**) PSi_5_ films at selected time points. DTX + 7-epi-DTX + 7-epi-10-oxo-DTX = 100%.

**Table 1 pharmaceutics-12-00444-t001:** Mechanical properties of blank and 1, 5 and 10% DTX-loaded polymer films.

Polymer Film	Ultimate Tensile Strength (MPa)	Elongation at Break (%)	Toughness (MJ m^−3^)	Young’s Modulus (kPa)
PEVA	3.63 ± 0.21	763 ± 21.9	1374 ± 11.6	10.3 ± 0.57
PEVA_1_	3.14 ± 0.27	732 ± 33.6	1263 ± 6.35	10.7 ± 1.52
PEVA_5_	2.64 ± 0.30	730 ± 30.0	1062 ± 6.35	10.7 ± 0.57
PEVA_10_	2.78 ± 0.16	642 ± 16.8	983 ± 11.0	11.7 ± 1.52
PU	11.3 ± 0.08	247 ± 22.4	1385 ± 13.5	199 ± 3.60
PU_1_	10.1 ± 0.14	237 ± 16.2	1479 ± 9.50	250 ± 5.00
PU_5_	8.49 ± 0.08	241 ± 29.8	968 ± 6.50	50.0 ± 3.46
PU_10_	8.47 ± 0.37	247 ± 21.0	892 ± 16.6	50.3 ± 5.50
PSi	2.50 ± 0.82	234 ± 22.3	214 ± 7.02	14.46 ± 2.63
PSi_1_	1.64 ± 0.12	169 ± 34.0	125 ± 3.51	23.60 ± 2.91
PSi_5_	1.62 ± 0.15	105 ± 7.96	65 ± 5.00	13.50 ± 0.27
PSi_10_	1.33 ± 0.34	88.9 ± 13.3	43 ± 6.02	12.38 ± 1.31

**Table 2 pharmaceutics-12-00444-t002:** Films parameters, theoretical and experimental loading, and percentage recovery of DTX from the PEVA, PU and PSi films (*n* = 3).

DTX-Loaded Polymer Film	Weight of the Film ^a^ ± SD (mg)	Film Thickness ± SD (µm)	Theoretical Loading ± SD (µg/cm^−2^)	Experimental Loading ± SD (µg/cm^−2^)	Percentage Recovery
PEVA_1_	30.02 ± 2.00	303.33 ± 2.89	475.29 ± 6.23	448.67 ± 9.97	94.4 ± 2.65
PEVA_5_	30.23 ± 0.92	306.67 ± 7.64	2476.5 ± 4.26	2387.4 ± 3.83	96.4 ± 0.32
PEVA_10_	31.77 ± 1.78	302.44 ± 5.08	5228.2 ± 8.30	4942.0 ± 12.28	94.5 ± 0.19
PU_1_	28.50 ± 1.58	296.67 ± 5.77	447.05 ± 0.95	411.92 ± 0.50	92.1 ± 0.93
PU_5_	29.78 ± 1.85	311.67 ± 10.41	2383.5 ± 4.72	2273.0 ± 4.13	95.4 ± 0.13
PU_10_	30.09 ± 2.48	302.45 ± 8.45	5031.8 ± 3.17	4614.1 ± 3.77	91.7 ± 0.21
PSi_1_	35.57 ± 3.26	316.67 ± 15.28	325.50 2.36	312.46 ± 6.36	96.0 ± 1.60
PSi_5_	36.67 ± 2.42	325.00 ± 13.23	2041.5 ± 1.91	1915.8 ± 11.58	93.8 ± 0.50
PSi_10_	38.04 ± 4.28	300.78 ± 12.54	3564.0 ± 3.99	3245.6 ± 10.16	91.1 ± 0.20

^a^ The area of the films: 120 ± 2.54 mm^2^.

**Table 3 pharmaceutics-12-00444-t003:** Correlation coefficient (R^2^) from drug release profile in Higuchi, first-order, zero-order, Hixon–Crowell and Korsmeyer–Peppa kinetic models.

Sample	Higuchi	First-Order	Zero-Order	Hixon–Crowell	Kors–Peppas
	(R^2^)	(R^2^)	(R^2^)	(R^2^)	(R^2^)
PEVA_5_	0.6271	0.5417	0.4069	0.493	0.966
PU_5_	0.9950	0.9613	0.9151	0.948	0.982
PSi_5_	0.9922	0.9672	0.9621	0.965	0.987

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
