# Peer review of "Influence of Polymer Composition on the Controlled Release of Docetaxel: A Comparison of Non-Degradable Polymer Films for Oesophageal Drug-Eluting Stents"

_pharmaceutics, 2020, doi:10.3390/pharmaceutics12050444_

Round 1
Reviewer 1 Report
Comments on a manuscript entitled: “Influence of polymer composition on the controlled release of docetaxel: a comparison of non-degradable polymer films for oesophageal drug-eluting stents” by Fouladian et al. submitted for publication in Pharmaceutics.
In this manuscript, an interesting study is described on the release of an anticancer drug from polymeric films intented to be coated onto oesophageal stents. The manuscript is written well and most (see below) conclusions are justified by the results. In my opinion, the manuscript can be accepted for publications after minor revisions.
In section 3.3 (XRPD measurements ), the authors claim that the drug was molecularly dispersed (line 263/4) based on the absence of typical sharp peaks of crystalline drug. However, the drug may also be dispersed in the polymer matrix as amorphous particles. In addition, the films contained 10 % drug. So, if for example 10-20 % of the drug would be crystalline, then 1-2 % of the total sample would be composed of crystalline drug. I think that XRPD is not sensitive enough to detect this? This could be elucidated by analyzing physical mixtures.
Also in section 3.5 (DSC measurements), the authors claim that the drug was moleclarly dispersed (line 307/8) based on the absence of the melting peak of the drug. Also here the drug could be present as amorphous particles (possibly, the change in Cp of the drug during its glass transition is too small to be detected). In addition, the drug melts a temperature which is a much higher than the melting or glass temperatuur of the polymers. Therefore, is it possible that the drug dissolves in the dissolved or rubbery polymer during the measurement. As a result, no melting enthalpy of the drug will be observed. Also these issues could be elucidated by analyzing physical mixtures.
Section 3.6: Is there a reason why incorporation of the drug into the PSi films has such a strong effect on the mechanical properties?
Section 3.7: in column entitled “Theoretical loading” SD values is missing. Using SD values of theoretical loading and SD values of experimental loading the SD values of percentage recovery could be calculated. It could appear that the recovery was not significantly decreased.
Section 3.8: Just for consideration: If indeed the drug was fully amorphous and molecularly dispersed in the matrix as the authors claim, it could crystallize in the matrix during incubation for 30 days in a buffer of 37oC, affecting the release profile.
Section 3.9: For readers not so familiar in the field, it would be appreciated to give the equations for the various release models (in the supplementary information). Is the R2 for PEVA in the first order model (0.5417) correct? Fit looks quite good! Please explain “n values” mentioned in line 497.
Author Response
Comments on a manuscript entitled: “Influence of polymer composition on the controlled release of docetaxel: a comparison of non-degradable polymer films for oesophageal drug-eluting stents” by Fouladian et al. submitted for publication in Pharmaceutics.
In this manuscript, an interesting study is described on the release of an anticancer drug from polymeric films intented to be coated onto oesophageal stents. The manuscript is written well and most (see below) conclusions are justified by the results. In my opinion, the manuscript can be accepted for publications after minor revisions.
In section 3.3 (XRPD measurements), the authors claim that the drug was molecularly dispersed (line 263/4) based on the absence of typical sharp peaks of crystalline drug. However, the drug may also be dispersed in the polymer matrix as amorphous particles. In addition, the films contained 10 % drug. So, if for example 10-20 % of the drug would be crystalline, then 1-2 % of the total sample would be composed of crystalline drug. I think that XRPD is not sensitive enough to detect this? This could be elucidated by analyzing physical mixtures.
We thank the reviewer for this comment. Yes, there is a possibility that the drug is dispersed into the polymer as amorphous particles because the sharp peaks are not visible, so we revised the explanation in the section 3.3 (lines 265-267).
In response to your suggestion about analysing the physical mixture of drug and polymers; due to the highly cross-linked nature of silicone and soft form of the other polymers having low Tgs, the physical blending of polymer and drug is not possible.
Also in section 3.5 (DSC measurements), the authors claim that the drug was molecularly dispersed (line 307/8) based on the absence of the melting peak of the drug. Also here the drug could be present as amorphous particles (possibly, the change in Cp of the drug during its glass transition is too small to be detected). In addition, the drug melts a temperature which is a much higher than the melting or glass temperatuur of the polymers. Therefore, is it possible that the drug dissolves in the dissolved or rubbery polymer during the measurement. As a result, no melting enthalpy of the drug will be observed. Also these issues could be elucidated by analysing physical mixtures.
As pointed out by the reviewer, the drug could be present as amorphous aggregates that lack a distinct melting point, or the drugs could dissolve in the polymer matrix when it melts; the description in section 3.5 of the manuscript has been updated to elaborate on this (Lines 310-314). Similar results were also reported by Mohsin et al. [1] with 20% DTX loading films; based on the absence of the drug’s melting peak in the drug loaded formulation, the authors claimed that the drug was dispersed in the system.
Section 3.6: Is there a reason why incorporation of the drug into the PSi films has such a strong effect on the mechanical properties?
We believe that the DTX may interfere with the cross-linking reaction or catalyst, leading to an overall reduction in cross-linking in silicone that leads to a reduction in the toughness; the description in section 3.6 of the manuscript has been updated to elaborate on this (please refer to line 416-418)
Section 3.7: in column entitled “Theoretical loading” SD values is missing. Using SD values of theoretical loading and SD values of experimental loading the SD values of percentage recovery could be calculated. It could appear that the recovery was not significantly decreased.
The SD values were added; refer to table 2.
Section 3.8: Just for consideration: If indeed the drug was fully amorphous and molecularly dispersed in the matrix as the authors claim, it could crystallize in the matrix during incubation for 30 days in a buffer of 37oC, affecting the release profile.
During experimentation, we did not anticipate this change and therefore, it was not monitored. This is an interesting prospect that we will consider trying to measure in future studies.
Section 3.9: For readers not so familiar in the field, it would be appreciated to give the equations for the various release models (in the supplementary information). Is the R2 for PEVA in the first order model (0.5417) correct? Fit looks quite good! Please explain “n values” mentioned in line 497.
We have included the equations in the supplementary information (Table S2). The R² value of the PEVA model (0.5417) is correct; the reason why the fit looks good is due to the logarithmic Y scale that was presented, which has a large range that compresses all the data. If plotted on a normal scale, the line of the best fit poorly fits to the data points.
Submission Date
16 April 2020
Date of this review
18 Apr 2020 13:11:50
[1] M. Shaikh, N. R. Choudhury, R. Knott, and S. Garg, “Engineering Stent Based Delivery System for Esophageal Cancer Using Docetaxel,” Molecular Pharmaceutics, vol. 12, no. 7, pp. 2305-2317, 2015/07/06, 2015.
Reviewer 2 Report
Manuscript: Influence of polymer composition on the controlled 2 release of docetaxel: a comparison of non-degradable 3 polymer films for oesophageal drug-eluting stents.
Reviewer
This article presents a very interesting approach of the study of drug eluting from several polymers for an application in the specific area of drug-eluting stents. However, some issues must be addressed before considering the article for publishing in Pharmaceuticals.
- Abstract, line 29: polymer-polymer interactions is not a very correct statement. The best was to substitute by “inter polymeric molecular interactions”, as polymer usually is associated with the all material.
- Throughout section 2 the number of replicates used in each characterization should be stated, as it is in section 2.8. Results from only one specimen, especially in polymers loaded with DTX, as no scientific validity, because the drug distribution can be heterogeneous.
- The SEM analysis could also present the cross-section of all the samples, with an EDS distribution of a chemical element specific to DTX. This way an image of such distribution could reinforce the statement made by the authors that ASSUME a homogenous distribution from the other characterization techniques.
- In sub-section 2.5 please present the wavelength of the CuKa radiation in nm. In the same sub-section, please specify the time per step.
- In sub-section 3.3, lines 262 to 264. Please revise this explanation. The fact that the diffraction peaks of DTX are not visible in the difractograms can have several explanations: the amount of DTX represents less than 5% of the volume that is being analyzed; the DTX is present in enough quantity but is amorphous, etc.
- In addition to the previous comment, it must be considered the DSC of the samples, because if the DTX is present in some amount, why does not the Tm of the drug does not appear (even very smoothed) in the polymer loaded samples.
- In section 3.6 I do not agree with the statement that the drug loading does not affect the mechanical properties of the Psi. It seems that the authors did not determined the Young’s modulus according to the recommended standards: at 0.2% or 1% strain. This implies that the initial portions of the stress-strain curves must be zoomed and the correct values of E determined. Besides, it is apparent from these curves and from the values of Table 1 that the DTX have an impact in the mechanical properties. Therefore this sub-section must be reconsidered and discussed with a different approach.
- For most of the manuscript the present results are of the polymers loaded with the higher degree of DTX. Nevertheless, the drug eluting profile of Figure 6 concerns the polymers loaded with 5% DTX and in the supplementary material the figure is of the polymers loaded with 1%. Please shoe the drug-eluting profile of all the polymers loaded with 10% DTX.
- The manuscript does not take in consideration the adhesion of the prepared drug-loaded polymers onto a metallic substrate that constitutes the stents. Therefore, even with interesting results the application of the developed systems to drug-eluting stents can be seriously compromised if the adhesion to the metallic substrate is bad. Therefore, I propose changing the title to “Influence of polymer composition on the controlled release of docetaxel: a comparison of non-degradable polymer films”
Author Response
Manuscript: Influence of polymer composition on the controlled 2 release of docetaxel: a comparison of non-degradable 3 polymer films for oesophageal drug-eluting stents.
This article presents a very interesting approach of the study of drug eluting from several polymers for an application in the specific area of drug-eluting stents. However, some issues must be addressed before considering the article for publishing in Pharmaceuticals.
Addressing the reviewer comments
Abstract, line 29: polymer-polymer interactions is not a very correct statement. The best was to substitute by “inter polymeric molecular interactions”, as polymer usually is associated with the all material.
The correction has been done, please refer to line 29 in the manuscript.
Throughout section 2 the number of replicates used in each characterization should be stated, as it is in section 2.8. Results from only one specimen, especially in polymers loaded with DTX, as no scientific validity, because the drug distribution can be heterogeneous.
The equipment used for DSC, TGA, PA Ft-IR and XRD in our university are validated and are routinely used for a large range of academic and commercial research studies. We have done most of these experiments in a single run because they were qualitative experiments and the results were consistent with the literature.
The SEM analysis could also present the cross-section of all the samples, with an EDS distribution of a chemical element specific to DTX. This way an image of such distribution could reinforce the statement made by the authors that ASSUME a homogenous distribution from the other characterization techniques.
We thank the reviewer for their thought on this. Whilst EDS may provide some indication of the drug distribution, it requires that there are distinct elements present in the drug and polymer, which is not always the case, e.g., all the elements in the polyurethane are also present in the drug. In addition, the spatial resolution of SEM EDS measurements is ~ 1 µm and therefore it would not tell us if the drug is molecularly dispersed or present as aggregates. The manuscript has been updated to remove claims to a homogenous distribution.
In sub-section 2.5 please present the wavelength of the CuKa radiation in nm. In the same sub-section, please specify the time per step.
The changes have been done, please refer to line 160-162 in the manuscript.
In sub-section 3.3, lines 262 to 264. Please revise this explanation. The fact that the diffraction peaks of DTX are not visible in the diffractograms can have several explanations: the amount of DTX represents less than 5% of the volume that is being analysed; the DTX is present in enough quantity but is amorphous, etc.
Yes there is a possibility that the drug is dispersed into the polymer as amorphous particles because the sharp peaks are not visible, so we revised the explanation in the section 3.3 (lines 265-268).
In addition to the previous comment, it must be considered the DSC of the samples, because if the DTX is present in some amount, why does not the Tm of the drug does not appear (even very smoothed) in the polymer loaded samples.
however, the DSC results confirmed that the drug cannot be presented as amorphous aggregates. In DSC analyses, if we assume that the drug transformed to the amorphous form, we expect to see a broader peak of DTX melting point in the formulation, nevertheless, no peak of DTX melting point was observed.
As discussed in response to reviewer 1, the DSC evidence would suggest that the drug is either molecularly dispersed in the polymer matrix, dissolves in the polymer matrix when it melts, of that the amorphous aggregates have such a broad range of melting points they are not observable in the DSC, even as a broad peak; the manuscript has been updated to elaborate on these possibilities. Please refer to section 3.5 (Lines 311-314).
In section 3.6 I do not agree with the statement that the drug loading does not affect the mechanical properties of the Psi. It seems that the authors did not determined the Young’s modulus according to the recommended standards: at 0.2% or 1% strain. This implies that the initial portions of the stress-strain curves must be zoomed, and the correct values of E determined. Besides, it is apparent from these curves and from the values of Table 1 that the DTX have an impact in the mechanical properties. Therefore, this sub-section must be reconsidered and discussed with a different approach.
In the section of 3.6 (lines 406-407) we claimed that, “the inclusion of DTX lead to a significant decrease in all measured parameters, even at low drug loadings (i.e., 1% w/w)”. and also in lines (416) we stated that “The significant decrease in the elongation at break for the PSi films upon DTX loading may be a problem if the film needs to be stretched significantly”. Hence, based on the results presented in the figure 5 and table 1, we strongly believe that the incorporation of the drug effects on the mechanical properties of PSi films. We considered your advice for determination of Young’s modulus at 1% strain and we checked the calculation again and made the correction. Please refer to table 1.
For most of the manuscript the present results are of the polymers loaded with the higher degree of DTX. Nevertheless, the drug eluting profile of Figure 6 concerns the polymers loaded with 5% DTX and in the supplementary material the figure is of the polymers loaded with 1%. Please shoe the drug-eluting profile of all the polymers loaded with 10% DTX.
We thank the reviewer for this point, and we have now added the release profiles for the 10% drug-loaded films in the SI, which are similar to the 5% loaded films. In section 3.10 (degradation studies) we only determined and presented the degradant products for 5% dug loadings for all polymers, so we decided to present the same concentration for release profile studies. We add an explanation in lines 490-492
The manuscript does not take in consideration the adhesion of the prepared drug-loaded polymers onto a metallic substrate that constitutes the stents. Therefore, even with interesting results the application of the developed systems to drug-eluting stents can be seriously compromised if the adhesion to the metallic substrate is bad. Therefore, I propose changing the title to “Influence of polymer composition on the controlled release of docetaxel: a comparison of non-degradable polymer films”
For the present study, we have nominated polysiloxane, polyurethane and PEVA polymers as they are widely used and common materials for covering metal stents commercially [1, 2]. Thus, there is already significant evidence that these polymers adhere to metal stents and are already used in this application. Furthermore, much of the testing of the polymer films was conducted with oesophageal cancer conditions in mind (e.g., release experiments in pH 6.5 media - a pH close to oesophagus lumen), and therefore the title highlights this focus of the research.
Submission Date
16 April 2020
Date of this review
23 Apr 2020 11:49:47
[1] P. Hindy, J. Hong, Y. Lam-Tsai, and F. Gress, “A comprehensive review of esophageal stents,” Gastroenterology & hepatology, vol. 8, no. 8, pp. 526-534, 2012.
[2] G. Nakazawa, A. V. Finn, F. D. Kolodgie, and R. Virmani, “A review of current devices and a look at new technology: drug-eluting stents,” Expert Review of Medical Devices, vol. 6, no. 1, pp. 33-42, 2009/01/01, 2009.
Reviewer 3 Report
The manuscript focus on analysis of Psi, PEVA and PU films for controlled release of docetaxel. I recommend to accept the manuscript after minor revision. The following points should be clarified:
- Drug release study – it has not been explained why only films with 5 % (Fig. 6) and 1 % (Fig. S2) of docetaxel were analyzed? Drug loading may influence the release properties and it would be very advantageous to compare matrices with 1, 5 and 10 % of drug content.
- Drug release mechanism: In the section 3.8 the drug release is described as: “…PEVA5 films provide a rapid dose followed by a relatively slow zero-order release, whereas PSi5 films provide a similar zero-order release without the initial burst release.”, which is not consistent with data presented in Table 3.
Author Response
The manuscript focus on analysis of Psi, PEVA and PU films for controlled release of docetaxel. I recommend to accept the manuscript after minor revision. The following points should be clarified:
Addressing the reviewer comments
Drug release study – it has not been explained why only films with 5 % (Fig. 6) and 1 % (Fig. S2) of docetaxel were analyzed? Drug loading may influence the release properties and it would be very advantageous to compare matrices with 1, 5 and 10 % of drug content.
As requested, the release profiles of the 10% drug loaded films has now been included in SI.
Drug release mechanism: In the section 3.8 the drug release is described as: “…PEVA5 films provide a rapid release followed by a relatively slow zero-order release, whereas PSi5 films provide a similar zero-order release without the initial burst release.”, which is not consistent with data presented in Table 3.
In section 3.8 for PEVA 5, we have mentioned “a rapid release followed by a relatively zero order” . the data in the table 3 is based on the overall release from day 0 to day 31. In section 3.8, when we talk about relatively slow zero order after a rapid release, we are not considering the day one that we have had a high amount of release (around 43%).
Submission Date
16 April 2020
Date of this review
23 Apr 2020 17:10:51
Round 2
Reviewer 2 Report
The reviewer appreciates the effort made by the authors to take in consideration the comments and suggestions.
Nevertheless, I steel have to underline my concern due to the fact that a number of characterizations techniques were only made without any replicates or repetitions. Independently of how calibrated and certified the equiments are, this does not imply heterogeneous behavior of the prepared materials.
For this reason I would like the Editoral team to decide if this particular point is important or not for the manuscript to be published.
All the other suggestions were correctly addressed by the authors.
Author Response
Thanks for your message.
There is no doubt that replicates in scientific experiments are important to validate the data. All the qualitative characterization studies performed in our lab are based on validated standard operating procedures. Before reporting the results, we ensure that the data is accurate and reproducible.
Having published over 200 peer-reviewed papers and acting as a regular reviewer including Guest editorship of the special issue in pharmaceutics, I differ from this reviewer’s comments. From our understanding of literature, it is not common to report the number of replicates in characterization studies, while that’s standard practice for studies like the release, drug loading, etc. Please refer to the following paper published in Pharmaceutics journal as well as the other good journals, you will note that none of these papers has reported the number of replicates for characterization studies.
[1] F. Afinjuomo, P. Fouladian, A. Parikh, T. G. Barclay, Y. Song, and S. Garg, “Preparation and Characterization of Oxidized Inulin Hydrogel for Controlled Drug Delivery,” vol. 11, no. 7, pp. 356, 2019.
[2] M. Rezvanian, M. C. I. M. Amin, and S.-F. Ng, “Development and physicochemical characterization of alginate composite film loaded with simvastatin as a potential wound dressing,” Carbohydrate Polymers, vol. 137, pp. 295-304, 2016/02/10/, 2016.
[3] P. Quan, X. Wan, Q. Tian, C. Liu, and L. Fang, “Dicarboxylic acid as a linker to improve the content of amorphous drug in drug-in-polymer film: Effects of molecular mobility, electrical conductivity and intermolecular interactions,” Journal of Controlled Release, vol. 317, pp. 142-153, 2020/01/10/, 2020.
[4] C. Zhang, X. Zhai, G. Zhao, F. Ren, and X. Leng, “Synthesis, characterization, and controlled release of selenium nanoparticles stabilized by chitosan of different molecular weights,” Carbohydrate Polymers, vol. 134, pp. 158-166, 2015/12/10/, 2015.
[5] F. Yu, Y. Li, Q. Chen, Y. He, H. Wang, L. Yang, S. Guo, Z. Meng, J. Cui, M. Xue, and X. D. Chen, “Monodisperse microparticles loaded with the self-assembled berberine-phospholipid complex-based phytosomes for improving oral bioavailability and enhancing hypoglycemic efficiency,” European Journal of Pharmaceutics and Biopharmaceutics, vol. 103, pp. 136-148, 2016/06/01/, 2016.
Hope, you will agree with my reasoning and accept our manuscript.
Regards
Sanjay Garg